# Clinical, laboratory, and hospital factors associated with preoperative complications in Peruvian older adults with hip fracture

Edwin Aguirre-Milachay[1], Darwin A. León-Figueroa[1], Mario J. Valladares-Garrido[2,3]*

1 Facultad de Medicina Humana, Universidad de San Martín de Porres, Chiclayo, Peru, 2 Universidad Continental, Lima, Peru, 3 Oficina de Inteligencia Sanitaria, Red Prestacional EsSalud Lambayeque, Chiclayo, Peru

* mvalladares@continental.edu.pe

**Data Availability Statement:** The data underlying the results presented in the study are available from https://doi.org/10.6084/m9.figshare.26507296.v1.

## Abstract

### Objectives

To determine the clinical, laboratory, and hospital factors associated with preoperative complications in older adults with hip fractures.

### Methodology

Analytical observational retrospective cohort study, whose population was older adults with a diagnosis of hip fracture treated in a hospital in northern Peru, during 2017–2019.

### Results

432 patients with a median age of 83 years (RIC: 77–88) were evaluated, with the female gender being the most prevalent (60.9%). The most common comorbidities included cardio-vascular disease (68%) and diabetes (17.6%), and multimorbidity was observed in 47.2% of cases. The median number of geriatric syndromes was 2 (RIC: 1–5). The overall mortality rate was 3.2% (1.7–5.3). Analysis with the Poisson regression model found a significant association with MRC scale 3–5 degree (RR = 1.60), glucose on admission (RR = 1.01), and minimally significantly female sex (RR = 2.41).

### Conclusions

The most commonly observed complications were infectious in nature, including pneumonia, sepsis, and urinary tract infections. The MRC scale from 3 to 5 degrees increases the risk of developing a preoperative complication; the glucose levels upon admission show a clinically irrelevant association; and in females, there is a minimally significant association in older adults with hip fractures.

**Funding:** The author(s) received no specific funding for this work.

**Competing interests:** The authors have declared that no competing interests exist.

## 1 Introduction

Osteoporotic hip fracture is the most common serious injury in the elderly and the major reason for hospital admission, anesthesia, and emergency surgery [1–3]. The incidence of hip fracture in the United States of America ranges from 78.7 to 195 per 100,000 people, with an in-hospital mortality rate of 2–5% and a one-year mortality rate of 20–24% [4]. In Spain, a 17-year registry found annual incidence rates in the over-65s of 767.8/100,000 (364.6 in men and 1087 in women) increasing exponentially with age up to 90 years [5]. According to the UK's National Hip Fracture Database (NHFD), hip fracture leads to a national hospital bed-day occupancy of one million, with only a minority recovering previous capabilities and a large proportion becoming increasingly dependent and limited in walking, requiring long-term care; this is associated with an annual cost of over $1.3 billion per year, which has led to strategic changes in its management in the UK healthcare system [6].

In Peru, a study on the clinical profile of hip fracture in a hospital of the armed forces found that it was more frequent in patients over 80 years of age, female, with 82% of comorbidity, the most frequent being arterial hypertension and diabetes [7].

The occurrence of complications in hip fracture patients is very frequent, with reports ranging from 17 to 49.6%, according to a systematic review [8]. Bielza et al. recorded 71.3% of complications in a Spanish population, the main ones being delirium (55.4%), renal failure (15.4%), and cardiac complications (12.3%) [9]; in contrast to data on the incidence of postoperative complications in a Japanese cohort of 11.2% (deep vein thrombosis 3.5%, pneumonia 1.8%, decubitus ulcers 1.6%) [10]. This is associated with an American Society of Anesthesiologists (ASA) III-IV classification and surgical delay [9].

In Latin America, according to data collected from a hospital in Mexico in 2016, the most frequent complications found in these patients were delirium (33%), pressure injuries, pneumopathy, and urinary retention, and only 33% had no complications [11]. Likewise, in Peru, it was found that in a National Hospital of the Ministry of Health, 62% had medical complications, and the median hospital stay for patients with surgical treatment was 26 days, clearly distant from the recommendations of the Clinical Guidelines, which recommend that this be done on the day of admission or one day after [12].

The majority of the studies speak of post-surgical complications because their protocols are optimized on the basis of Clinical Practice Guidelines (CPG), standards of care and quality indicators, short preoperative times, and multidisciplinary management in orthogeriatric units [13, 14]. In Latin American studies, however, there are few orthogeriatric units, and even the adaptation of CPG, or management standards, is scarce or nonexistent [14]. This leads to a higher prevalence of pre-surgical complications due to prolonged hospital stays. Furthermore, no evidence has been found to evaluate the possible factors influencing the presence or development of these complications.

The aim of this research is to determine the clinical, laboratory, and hospital factors associated with preoperative complications in older adults with hip fractures. Knowledge of these data in this population will help to understand their vulnerability to the development of preoperative complications, including analysis of the hospital and prehospital data that may have an impact on the presence of these complications, which in sum bring high costs to our health system. Given the still-insufficient adaptation of our health system to international standards, this study could contribute to prioritizing surgical interventions in this population group.

## 2 Material and methods

### 2.1 Study design

An analytical observational retrospective cohort study was conducted, focusing on the population of older adults aged 60 years or older with a diagnosis of hip fracture, attended at the Almanzor Aguinaga Asenjo National Hospital (HNAAA) in the Lambayeque region, Peru, during the period from October 2017 to April 2019.

### 2.2 Eligibility criteria

Older adult patients with a diagnosis of hip fracture, whether cervical, intertrochanteric, or subtrochanteric, were included in the study. In addition, those elderly adults who underwent surgical intervention within the period established by the corresponding medical service were considered, as well as those who had received care from the Orthopedic Geriatrics unit of the HNAAA. Excluded from the study were older adult patients with pathological hip fractures due to cancer or osteomalacia, those with multiple fractures, voluntary discharge, and those whose collection forms did not contain all the data on the variables under study at the time the research was carried out. This excluded population did not require further analysis as it was not considered to affect the results.

The unit of analysis was considered to be the patient with one or more pre-surgical complications.

### 2.3 Sample and sampling

The sample was obtained using the online calculator from the Cleveland Clinic (https://riskcalc.org/samplesize/), determined based on an exposed proportion of 14% and a non-exposed proportion of 0.05% for a perioperative complication such as pneumonia, taking the variable delayed surgery with a cutoff point of 48 hours, based on a systematic review by Klestil et al. [15], with a non-exposed/exposed ratio of 1 (considering that the only evidence of the prevalence of pre-surgical complications in the Peruvian population was 62%), a confidence level of 95%, a power of 80%, and a percentage loss of 20%. The sample obtained was 414.

Non-random, purposive sampling was carried out. The unit of analysis was the medical records and data collection form of older adult patients with hip fractures treated at the HNAAA.

### 2.4 Study variables

The dependent variable is defined as the preoperative complication, which refers to any medical complication that occurs before surgery and is evaluated from the time of admission to the emergency room. Complications were classified as majors: sepsis, pneumonia, stroke, gastrointestinal bleeding, partial or total intestinal obstruction, and minors: urinary tract infection without sepsis, pressure ulcer, and delirium.

In addition, the main independent variables were defined as follows: a) Emergency stay: refers to the time the patient spent in the emergency area before being hospitalized, b) preoperative time: it refers to the time in days from admission to the emergency room until surgery, c) Multimorbidity is defined as the presence of two or more comorbidities in the patient, d) Geriatric syndromes: refers to the main geriatric syndrome present before hospital admission, e) FAC scale (Functional Ambulation Category): is a category that evaluates the patient's ambulatory capacity, with a score ranging from 0 to 5. f) "Mental Red Cross" scale (MRC): refers to an index that evaluates the patient's cognitive status, which was validated by the Red Cross hospital in Madrid [16], with a score ranging from 0 to 5 degrees from completely normal = 0, slight disturbances of disorientation in time, he/she maintains a conversation correctly = 1,

disorientation in time, the conversation is possible but not perfect. He knows people well, even though he sometimes forgets something. Personality disorders. Occasional incontinence = 2, disorientation, it´s impossible to have a logical conversation: it confuses people, clear mood disorders, frequent incontinence = 3, disorientation, clear mental alterations, habitual or total incontinence = 4 to advanced dementia with a vegetative state, with or without episodes of agitation, and total incontinence = 5 [17]. g) Barthel Index: an index that evaluates the patient's activities of daily living, with a score ranging from 0 to 100. h) Traumatological diagnosis refers to the type of fracture defined by the traumatologist.

## 2.5 Procedures and techniques

A record and observation were made of the ortho-geriatric assessment form and the logbook kept by the ortho-geriatrics functional unit of the HNAAA from October 2017 to April 2019. The data collection form for geriatric patients with hip fractures consists of 7 sections: affiliation, hospital stay, preoperative time, geriatric comorbidities and syndromes, baseline functional and cognitive status, laboratory data, trauma diagnosis, and complications. This collection form was elaborated with the advice of geriatric doctors belonging to the orthogeriatric unit, with its qualitative validation by expert judgement.

The research was conducted using a protocol approved by an institutional ethics committee. Permission for data collection was requested from the relevant departments and service heads. Data collection started on January 6, 2022, and continued until March 28, 2022, using the data collection form with the patient record book of the orthogeriatric unit. From the target population, the accessible population was determined, and from this, the sample was obtained. The corresponding exclusion and elimination of units of analysis that fulfilled the selection criteria were carried out. The information pertaining to the units of analysis was entered into a database, describing whether they presented a perioperative complication and the type of this complication, which was corroborated in the clinical history, in addition to the hospital times, which were corroborated in the registration notebooks and the laboratory times in the management system. The registration in the database was carried out by two people independently and later compared to avoid errors in the registration of information.

A hospital management protocol for hip fracture has not been established; therefore, no process or outcome indicators are determined for the management of this condition.

## 2.6 Data analysis plan

The analysis was carried out using the Stata software package, version 17. Univariate, bivariate, and multivariate analyses were performed.

Univariate analysis: Measures of central tendency and dispersion were performed for quantitative variables according to normality criteria, as well as frequencies and percentages for qualitative variables not related to a specific variable according to epidemiological, clinical, and laboratory characteristics.

These same measures were determined in patients with and without perioperative complications in a base table.

Bivariate analysis: Parametric data were analyzed using independent samples. Student's t-tests for quantitative and quanti-qualitative variables, Chi-square for nominal qualitative variables, and Mann-Whitney U for ordinal variables are based on a $p<0.05$ (95% confidence level). Measures of effect or strength of association were measured as relative risk (RR) with their corresponding confidence intervals between the dependent variable and the intervening variables showing association in the bivariate analysis.

Multivariate analysis: A Poisson regression model with robust variance was performed, with adjustments for clinical factors such as cognitive status, functional status, comorbidities, and type of fracture; laboratory factors such as hemoglobin; and hospital factors such as time spent in the emergency room and time of arrival at the emergency room. In addition to evaluating assumptions such as collinearity in the adjusted analysis.

### 2.7 Ethical considerations

The present study had the permission of the Training Unit of the HNAAA. It also had the approval of the Ethics Committee of the Hospital Almanzor Aguinaga Asenjo of the Lambayeque Health Network, whose code is CIEI-RPL: 064-DIC-2021.

The confidentiality of the data was determined by the use of self-generated codes produced by the Essalud (Social Health Security) management system in identifying the patients who registered in the collection form and were subsequently uploaded to a database. Esto está garantizado por el Artículo 25 del Capítulo I, Título II de la Ley General de Salud. The data remained encoded by the system's autogenerated codes that replaced personal identification; these remained in a single-use database for the researchers, and after their analysis and publication, they will be deleted within a maximum period of 2 years.

The exemption from informed consent was requested due to the minimal risk to participants as randomization has not been carried out and the changes in care compared to the standard of care. Additionally, the study is designed to compare the care provided against predetermined standards [18] and because we work with data recorded by the Geriatric Orthopedics Unit of the HNAAA.

## 3 Results

The initial sample consisted of 439 older adults with hip fractures, after excluding 7 for pathological fracture, voluntary discharge, and multiple fractures; a final sample of 432 older adult hip fracture patients was obtained (Fig 1). This sample was taken because it was larger than the sample size measured (414).

### 3.1 General population characteristics and multivariate analysis

The epidemiological data are shown in Table 1. With respect to age, we see a minimum of 60 years and a maximum of 100 years; the most frequent age range was 80 to 100 years (60.88%), and the least frequent was 60 to 79 years (17.59%). An association was found between female sex and the presence of surgical complications ($p \leq 0.02$).

Regarding waiting times, the maximum waiting time to go to the emergency room was 45 days, and the maximum length of stay in the emergency room before surgery was 26 days, the minimum preoperative time was 3 days, and the maximum was 41; an association was found between this variable and preoperative complications ($p < 0.0001$).

The most frequent comorbidity was cardiovascular disease, including arterial hypertension, arrhythmias, heart disease, and peripheral arterial disease. The least frequent was liver disease, such as liver cirrhosis. An association was found with the presence of comorbidities ($p \leq 0.004$) and with multimorbidity ($p \leq 0.003$).

Within the geriatric assessment evaluated, moderate-severe dependency as measured by the Barthel index was present in 93 (21.53%), mild dependency in 76 (17.59%), and independence in basic activities of daily living was the most frequent with 263 (60.88%). In addition, we found, according to the FAC (Functional Ambulation Classification) scale, that up to 119 (37.53%) walked without help or supervision, and 72 (30.64%) did not walk or required the help of two people. No statistical association was found with any functional assessment scale.

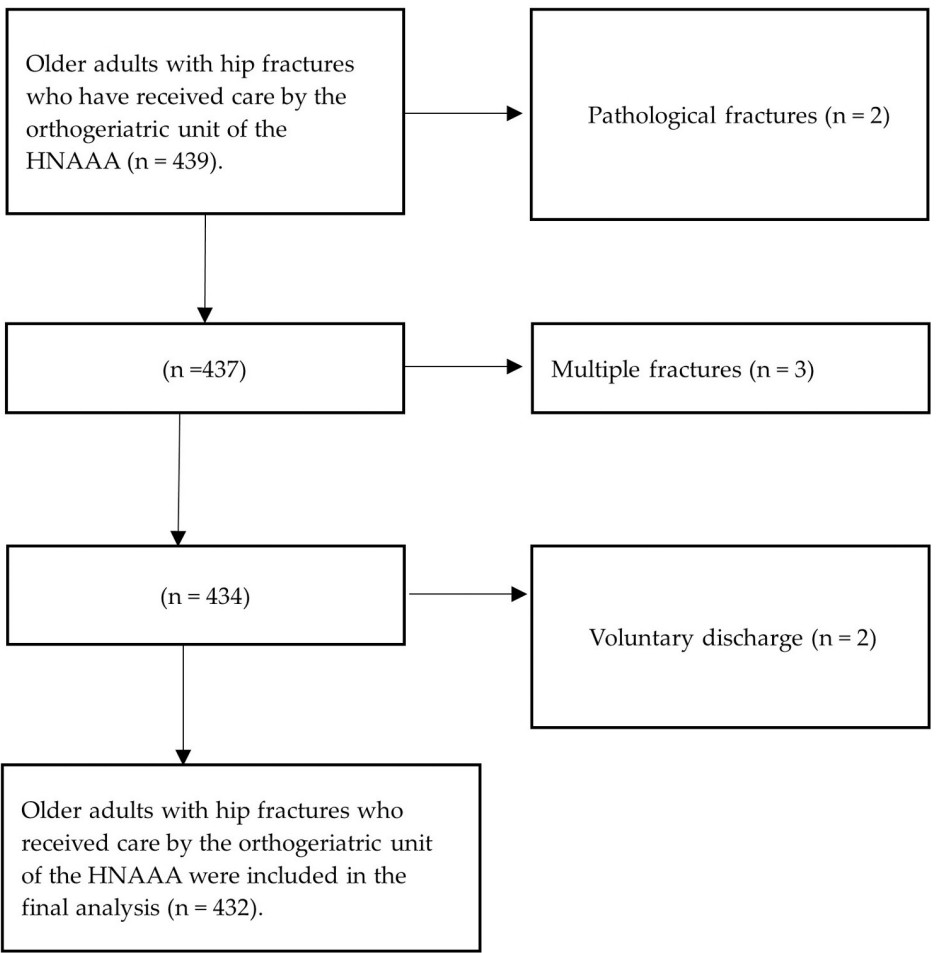

**Fig 1. Participant selection flowchart.**

It was also determined according to the Mental Red Cross (MRC) scale that 30.64% had no cognitive impairment, the most frequent category being those with mild problems (48.09%) and severe problems (21.18%). No significant association was found with pre-surgical complications.

The most frequent geriatric syndromes found were dementia and depression. An association was found between the number of geriatric syndromes and pre-surgical complications ($p \leq 0.01$).

With respect to laboratory tests, we found minimum hemoglobin values on admission of 4.6 g/dl, minimum platelet values of up to 29,000, maximum INR (International Normalized Ratio) values of 12.8, and maximum urea and creatinine values of 191 and 14 mg/dl, respectively. Only a minimally significant association was found with glycemia on admission ($p \leq 0.007$), with minimum values of 15 and maximum values of 334 mg/dl. 14 older adults died during the study period, with a statistical association found with pre-surgical complications ($< 0.0001$).

We found missing data in variables such as sex, FAC scale, number of geriatric syndromes, MRC scale, glucose, hemoglobin, serum creatinine, and multimorbidity. The categorical and numerical variables that were found to be associated in the bivariate analysis received simple imputation for the subsequent raw analysis using Poisson regression.

**Table 1. General characteristics according to pre-surgical complications.**

| Variables | Total | Complications | | p-value |
|---|---|---|---|---|
| | N = 432 | Yes = 99 | No = 333 | |
| Age (years) | 83 (77–88)** | 84 (77–89)** | 83 (77–88)** | 0.207¥ |
| Sex ⁂ | | | | |
| Male | 151 (35.3) | 21 (13.91) | 130 (86.09) | **0.02¥** |
| Female | 277 (64.7) | 74 (26.71) | 203 (73.29) | |
| Time from fall to emergency (days) | 1 (0–1)** | 0 (0–1)** | 1 (0–1)** | 0.88¥ |
| Emergency stay (days) | 3 (2–5)** | 3 (1–5)** | 3 (2–4.5)** | 0.95¥ |
| Preoperative time (days)⁂ | 14 (10–19) | 17 (13–24) | 13 (10–17) | **<0.0001** |
| **Comorbidities:** | | 2 (1–3)** | 1 (1–2)** | **0.004** |
| Cardiovascular disease | 81 (68) | 18 (22.2) | 63 (77.8) | |
| Diabetes mellitus | 21 (17.6) | 7 (33.3) | 14 (66.6) | |
| Cerebrovascular Disease | 10 (8.4) | 5 (50) | 5 (50) | |
| Osteoarticular disease | 17 (14.2) | 3 (17.6) | 14 (82.4) | |
| Neurological disease | 14 (11.7) | 6 (42.9) | 8 (57.1) | |
| Liver Disease | 6 (5) | 3 (50) | 3 (50) | |
| Chronic Kidney Disease | 10 (8.4) | 3 (30) | 7 (70) | |
| Cancer | 10 (8.4) | 2 (20) | 8 (80) | |
| >1 comorbidities⁂ | 102 (47.2) | 36 (35.29) | 66 (64.71) | **0.003¥** |
| **Geriatric Assessment** ⁂ | | | | |
| Barthel Index | 90 (70–100) | 90 (60–100) | 90 (70–100) | 0.19¥ |
| Cognitive dysfunction scale (Mental Red Cross) ⁂ | | 2 (1–3) | 1 (0–2) | **0.09¥** |
| MRC scale 3–5 degree | 50 (21.3) | 14 (28) | 36 (72) | 0.13 |
| Functional Ambulation Classification (FAC) ⁂ | 2.72 ±1,64* | 2.72±1.64* | 2.78±1.64* | 0.77 |
| Geriatric Syndromes⁂ | 2 (1–5) | 3 (2–6) | 2 (1–5) | **0.01¥** |
| Dementia | 36 (53.7) | 7 (19.4) | 29 (80.6) | 0.83 |
| Visual deprivation | 10 (14.9) | 3 (30) | 7 (70) | |
| Previous falls | 8 (11.9) | 2 (25) | 6 (75) | |
| Depression | 13 (19.4) | 2 (15.4) | 11 (84.6) | |
| **Laboratory tests (hospital admission)** ⁂ | | | | |
| Hemoglobin (g/dl) | 10.95±1.8* | 10.75±1.81* | 11.01±1.89* | 0.42¥ |
| Leukocytes | 10040 (7960–12360)** | 10425 (9050–12900)** | 9940 (7790–12350)** | 0.29¥ |
| Platelets x 10³ | 234 (182–290)** | 233 (177.5–261.5)** | 235 (186–297)** | 0.57 |
| Lymphocytes (%) | 13 (9–17)** | 13.5 (7–18.5)** | 13(9–17)** | 0.78¥ |
| Glucose (mg/dl) | 122 (104–150)** | 135 (106–152.5)** | 119 (104–144)** | **0.07¥** |
| Urea (mg/dl) | 41 (30–54) ** | 43.5 (29–65)** | 40 (30–51)** | 0.43¥ |
| Creatinine (mg/dl) | 0.76 (0.6–1) ** | 0.82 (0.61–1.18) ** | 0.75 (0.57–0.985) ** | 0.11¥ |
| INR | 1.02 (0.97–1.09) ** | 1.03 (0.97–1.1) ** | 1.02 (0.98–1.08) ** | 0.78¥ |
| **Traumatological Diagnosis** | | | | |
| Femoral neck fracture | 155 (29.5) | 27 (23.48) | 88 (76.52) | 0.94¥ |
| Intertrochanteric fracture | 234 (60) | 51 (21.79) | 183 (78.21) | |
| Subtrochanteric fracture | 41 (10.5) | 9 (21.95) | 32 (78.05) | |
| **Mortality** | 14 (3.2) | 10 (71.4) | 4 (28.6) | **<0.0001** |

\* Mean and standard deviation,

\*\* Median and interquartile ranges (25–75%),

¥ P-values calculated using the Chi² test, Fisher's exact test, and Mann-Whitney U test, as applicable. Mental Red Cross (MRC), Functional Ambulation Classification (FAC), and International Normalized Ratio (INR).

⁂ There are variables such as sex (4 patients), geriatric assessment variables (Barthel index, MRC, and FAC), and analytical variables that have missing data.

**Table 2. Types of pre-surgical complication.**

| Variables | Total |
|---|---|
| | n = 116; 100% |
| **Majors** | |
| Sepsis | 11 (9.4) |
| Pneumonia | 32 (27.6) |
| Stroke | 3 (2.6) |
| Gastrointestinal bleeding | 3 (2.6) |
| Partial or total intestinal obstruction | 5 (4.3) |
| **Minors** | |
| Urinary tract infection | 41 (35.3) |
| Pressure ulcer | 9 (7.8) |
| Delirium | 12 (10.3) |

### 3.2 Frequency and type of pre-surgical complications

Pre-surgical complications are reported in Table 2. 116 pre-surgical complications were detailed in 99 patients; the most frequent major complication was pneumonia, and the most frequent minor complication was a urinary tract infection.

### 3.3 Multivariate analysis

In the regression model, the Poisson model with robust variance was used due to the prevalence of complications exceeding 10%, which guided us to use RR. A crude analysis was performed between the presence of pre-surgical complications and variables such as age over 80 years, sex, multimorbidity, Barthel index less than 60 points, MRC scale, number of geriatric syndromes, glucose on admission, hemoglobin <10 g/dl, and preoperative time.

In the multivariate analysis, we conducted prediction models using demographic variables (sex), epidemiological variables (multimorbidity, number of geriatric syndromes, MRC scale), analytical variables (glucose at admission), and hospital variables (preoperative time) that were significantly associated in the crude analysis or where the univariate model in that analysis had a p-value <0.20, following the confounder adjustment model with the univariate analysis.

Multicollinearity was assessed in the multivariate analysis, finding in the final model an average VIF of 2.76 (sex = 2.41, multimorbidity = 2.52, CRM scale 3–5 degree = 1.39, number of geriatric syndromes = 3.94, and glucose at admission = 3.56). Preoperative time was excluded due to having a high VIF of 6.83 and altering the balance of the model.

The model found that an MRC scale of > = 3 degree RR = 2.16 (1.13–4.14) and admission glucose RR = 1.01 (1.001–1.012) increased the risk of developing preoperative complications, and minimally significant female sex RR = 2.43 (1.02–5.80) showed an association with preoperative complications in older adults with hip fractures (p = 0.045) (Table 3).

## 4 Discussion

Hip fracture is the most common serious injury in the elderly and the one that causes the greatest functional impairment and loss of mobility in the short and long term [6]. It is also associated with the occurrence of post-surgical morbidities [19] and, to a lesser extent, hospital mortality, although hospital mortality may have increased during the years 2020–2021 due to the COVID-19 pandemic [20]. However, there are few studies that evaluate complications prior to surgery in this population. This is because there is an early surgical treatment standard

**Table 3. Poisson regression.**

| Variable | RR | P-value | RR adjusted | P-value | 95% CI |
|---|---|---|---|---|---|
| Age >80 years | 0.97 | 0.911 | - | - | - |
| Sex | | | | | |
| Male | Ref. | | Ref. | | |
| Female | 1.92 | 0.004 | **2.43** | **0.045** | 1.02–5.80 |
| Multimorbidity | 2.01 | 0.004 | 1.60 | 0.251 | 0.71–3.59 |
| Barthel index <60points | 1.12 | 0.618 | - | - | - |
| MRC scale 3-5degree | 1.52 | 0.126 | **2.16** | **0.019** | 1.13–4.14 |
| Number of geriatric syndromes | 1.14 | 0.004 | 1.04 | 0.607 | 0.89–1.21 |
| Glucose on admission | 1.004 | 0.024 | **1.01** | **0.036** | 1.001–1.012 |
| Hemoglobin <10gr/dl | 0.74 | 0.277 | - | - | - |
| Preoperative time | 1.606 | <0.0001 | - | - | - |

Relative Risk (RR), Mental Red Cross (MRC), and 95% confidence interval (CI)

of 36 hours established by the National Institute for Health and Care Excellence (NICE) in 2014 [6, 21] that is not adequately met in our country due to the current socio-sanitary condition [12].

## 4.1 Prevalence of preoperative complications in older adults with hip fractures

Preoperative complications occurred in 26.9% of adults. This prevalence is lower than that reported by Palomino in Lima with 62% [12] and Barrios in Mexico with 67% [11]. Additionally, the reported complications are mostly post-surgical, as the Spanish study by Bielsa found 71% [9], the Spanish registries in 2019 considered the prevalence of pressure injuries at 7.2% [14], and the English NHFD registry found 38% of post-surgical delirium in 2022 [13]. This prevalence could be explained by under-reporting of these complications due to prolonged emergency waiting time and especially under-reporting of delirium as a pre-surgical complication.

Among the main pre-surgical complications, infectious complications are the most frequent. Pneumonia and sepsis were the most frequent major complications, and urinary tract infections were the most frequent minor ones. The etiology of sepsis was not detailed. Similar results were found in a study from the United Kingdom, where the complications, in this case post-surgical, were infectious, followed by cardiovascular complications. The diagnosis of delirium was less frequent at 7.6% than in our study with 10.3% [19]. However, only septic complications were associated with delayed surgery [22].

## 4.2 Clinical factors are associated with preoperative complications in older adults with hip fractures

Our study found an increased risk of developing a surgical complication in patients with MRC 3–5 degree, glucose on admission, and a tendency towards association in female sex in older adults with hip fracture.

Age and gender have been evaluated as predictors of mortality and post-surgical complications in several studies. A study in the USA considered male sex and especially age as predictors of mortality [23], as did a study in Japan where age over 90 years was associated with major postoperative complications overall [24]. Higashikawa, in another study with a Japanese

population, attributed female sex as a predictor of aspiration pneumonia [25]. Although our study did not find an association between age and the presence of complications in bivariate and multivariate analyses, a marginally significant association was found between female sex and preoperative complications. This may be due to a higher distribution of females between the ages of 60 and 90 in our population and in other studies, as well as a greater prevalence of extracapsular fractures in that gender [26], which is also related to age and could be associated with increased complications such as subtrochanteric fractures [27]. Although this has not been evaluated in this study, one could assume a potentially higher prevalence of frailty related to age and gender distribution, which is associated with postoperative delirium [28].

The presence of comorbidities impacts the development of post-surgical complications. Chronic kidney disease has been associated with mortality from post-surgical pneumonia, with follow-ups up to 8 months [29]. Chronic obstructive pulmonary disease, heart failure, and advanced cancer also appear to increase the risk of major complications (pulmonary, renal, cardiac, and sepsis) [22]. Although a high body mass index (BMI) has not been associated with complications, it may increase preoperative time and hospital stay [30], with the exception of BMI greater than 40, where increased respiratory and dermal complications have been reported [31].

Although our study only found a clinical association with multimorbidity in the raw analysis, an Italian study, which evaluated in-hospital mortality, found that multimorbidity was associated with this event [32]. Furthermore, the population with hip fractures has a high prevalence of multimorbidity, with cardiac and pulmonary disorders predominating, and a one-year mortality risk measured at 8% according to the study by Lloyd et al. [33]. Conditions such as diabetes are considered independent mortality factors within a year for patients with hip fractures [34]. This difference is likely due to the importance of establishing an adequate assessment of multimorbidity using indices like the modified Charlson, which are associated with adverse events in this population, particularly mortality, but they are also associated with decision-making and an increase in preoperative time [35].

Our study found an association between the number of geriatric syndromes present in each patient and pre-surgical complications, without finding a clinical association in the adjusted model or a specific association with depression, visual deprivation, dependence, or mobility. We also found a raw but unadjusted association between functional dependency measured by the Barthel index and preoperative complications. However, our study finds that a mental red cross (MRC) scale 3–5 degree equating to moderate to very severe dementia increased the risk of pre-surgical complications by more than 4-fold. Several studies have assessed geriatric syndromes such as cognitive impairment and disability in hip fracture patients. A Japanese study found that a history of cognitive impairment and dependence on basic activities, as measured by the Barthel index, can predict the occurrence of aspiration pneumonia [25]. Functional dependence and impaired mobility have also been associated with increased mortality at 30 days and 3 months, respectively [22, 32], but not with preoperative complications. Yao W et al., in a systematic review, found an association between functional dependence and cognitive impairment with postoperative pneumonia, but with a high degree of heterogeneity among the studies [36], and Mosk et al., in a Dutch population study, found that dementia and a previous history of delirium were predictors of perioperative delirium and that the presence of dementia was associated with more than one complication [37]. This association could be due to the fact that, although delirium was the third most frequent preoperative complication found in our study, the presence of dementia increases the risk of developing delirium by being a predisposing factor and making these patients more vulnerable even to other complications due to their poor nutritional status and functional dependence.

Our study found that preoperative time was associated in the raw analysis with pre-surgical complications, but we were unable to confirm its significance in the adjusted analysis. Several studies highlight the importance of preoperative time in the outcomes of patients with hip fractures, such as hospital stay, quality of life, and readmission [38]. Additionally, a time to surgery greater than 48 hours is associated with hospital pneumonia; the incidence of hospital pneumonia increased by 1.09 times [36]. Additionally, a preoperative time within 48 hours reduces the risk of mortality by 20% within a year. It was also found that a preoperative time greater than 48 hours was associated with pressure injuries, regardless of severity, and with urinary infections, without distinguishing whether they were related to the presence of a catheter [39, 40]. Furthermore, associations were found with venous thrombosis or stroke [15, 38]. In contrast to the study by Varady N. et al., which found that in a population in the U.S. with pathological hip fractures, a delay in surgery of more than 2 days was associated with an increased length of hospital stay but not with postoperative complications [22]. It should be noted that most studies focus on postoperative complications, and although it is not possible to define a specific complication related to delayed preoperative time in systematic reviews, there is a trend towards a greater association with pneumonia and pressure injuries. This could not be detailed in the present study and requires further evaluation of these conditions.

### 4.3. Laboratory factors are associated with preoperative complications in older adults with hip fractures

Laboratory parameters may play a role in the development of pre- and post-surgical complications. We found a significant association between admission blood glucose levels and the risk of developing preoperative complications. Although it has been shown that glucose levels at admission are associated with a moderate risk of postoperative pneumonia in patients with hip fractures, and higher values increase the likelihood of this event, no studies have been found regarding preoperative complications. While our study guides us to evaluate this laboratory data, its clinical relevance is minimal in our research [41]. We found no association with other laboratory results; however, other authors, such as Shuai, determined that a neutrophil/lymphocyte index greater than 4.85 is related to post-surgical delirium [42] or Wang Y. et al. found a higher risk of postoperative pneumonia in patients with hypoalbuminemia [43].

### 4.4 Implications of the findings for geriatrics and public health

The assessment of preoperative and perioperative complications is of great importance in predicting adverse health outcomes in older adults with hip fractures. Complications such as delirium play an important role because they are associated not only with increased discharge to rehabilitation units or readmissions but also with mortality at 30 and especially 180 days after discharge [44].

### 4.5 Limitations and strengths

Our study has limitations such as the lack of a standardized definition of complications such as delirium and urinary tract infection, the loss of records of the requested analytical tests, which hinders their proper assessment, as well as a lower proportion of geriatric syndromes and cognitive impairment and mobility test scores. In addition, there is a selection bias, given that the sampling used was non-probabilistic and intentional, and it is not possible to infer the findings from the entire population of interest. Additionally, it was not possible to assess other confounding variables potentially associated with preoperative complications in older adults, such as ASA and frailty assessment, that could influence the results [45].

However, this research has strengths such as the large study population and the follow-up given to the patients during their hospital stay by the orthogeriatric team, which was created in 2017, which facilitated obtaining the outcome variables and measuring geriatric variables adequately.

## 5 Conclusions

The most frequent complications were infectious complications such as pneumonia, sepsis, and urinary tract infections. The MRC scale from 3 to 5 degrees increases the risk of developing a pre-operative complication; the glucose levels upon admission show a clinically irrelevant association; and in females, there is a minimally significant association in older adults with hip fractures.

Our study allows us to obtain valuable information on pre-surgical complications in a population of older adults with hip fractures who have long hospital stays and high pre-surgical times in our health system, and additionally, it allows us to recognize that population is at higher risk of presenting these complications. This will allow us to better evaluate and prioritize the surgical management of this population while establishing national programs with appropriate indicators for this population.

Additionally, studies related to post-surgical complications, functionality, and post-hospital mortality are recommended due to the large amount of international information that indicates that this population group presents negative outcomes in the short and medium term with high consumption of health resources.

## Supporting information

**S1 Checklist. STROBE statement—Checklist of items that should be included in reports of observational studies.**
(PDF)

## Author Contributions

**Conceptualization:** Edwin Aguirre-Milachay, Darwin A. León-Figueroa, Mario J. Valladares-Garrido.

**Data curation:** Edwin Aguirre-Milachay, Darwin A. León-Figueroa, Mario J. Valladares-Garrido.

**Formal analysis:** Edwin Aguirre-Milachay, Darwin A. León-Figueroa, Mario J. Valladares-Garrido.

**Investigation:** Edwin Aguirre-Milachay, Darwin A. León-Figueroa, Mario J. Valladares-Garrido.

**Methodology:** Edwin Aguirre-Milachay, Darwin A. León-Figueroa, Mario J. Valladares-Garrido.

**Project administration:** Edwin Aguirre-Milachay, Mario J. Valladares-Garrido.

**Resources:** Edwin Aguirre-Milachay, Darwin A. León-Figueroa, Mario J. Valladares-Garrido.

**Software:** Edwin Aguirre-Milachay, Darwin A. León-Figueroa.

**Supervision:** Edwin Aguirre-Milachay, Darwin A. León-Figueroa, Mario J. Valladares-Garrido.

**Validation:** Edwin Aguirre-Milachay, Darwin A. León-Figueroa.

**Visualization:** Edwin Aguirre-Milachay, Mario J. Valladares-Garrido.

**Writing – original draft:** Edwin Aguirre-Milachay, Darwin A. León-Figueroa, Mario J. Valladares-Garrido.

**Writing – review & editing:** Edwin Aguirre-Milachay, Darwin A. León-Figueroa, Mario J. Valladares-Garrido.

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
