## [Decision Letter · Decision Letter 0]

15 Sep 2024

PONE-D-24-33388Clinical, laboratory, and hospital factors associated with preoperative complications in Peruvian older adults with hip fracture.PLOS ONE

Dear Dr. Valladares-Garrido,

Thank you for submitting your manuscript to PLOS ONE. After careful consideration, we feel that it has merit but does not fully meet PLOS ONE’s publication criteria as it currently stands. Therefore, we invite you to submit a revised version of the manuscript that addresses the points raised during the review process.

We look forward to receiving your revised manuscript.

Kind regards,

Barry Kweh

Academic Editor

PLOS ONE

Journal Requirements: When submitting your revision, we need you to address these additional requirements. 1. Please ensure that your manuscript meets PLOS ONE's style requirements, including those for file naming. The PLOS ONE style templates can be found at https://journals.plos.org/plosone/s/file?id=wjVg/PLOSOne_formatting_sample_main_body.pdf and https://journals.plos.org/plosone/s/file?id=ba62/PLOSOne_formatting_sample_title_authors_affiliations.pdf 2. Please include captions for your Supporting Information files at the end of your manuscript, and update any in-text citations to match accordingly. Please see our Supporting Information guidelines for more information: http://journals.plos.org/plosone/s/supporting-information. 3. Please review your reference list to ensure that it is complete and correct. If you have cited papers that have been retracted, please include the rationale for doing so in the manuscript text, or remove these references and replace them with relevant current references. Any changes to the reference list should be mentioned in the rebuttal letter that accompanies your revised manuscript. If you need to cite a retracted article, indicate the article’s retracted status in the References list and also include a citation and full reference for the retraction notice.

Additional Editor Comments:

A well written article which requires clarification of the methodology and a broader overview of the literature including alternate means of risk stratification peri-operatively.

Reviewers' comments:

Reviewer's Responses to Questions

**Comments to the Author**

1. Is the manuscript technically sound, and do the data support the conclusions?

Reviewer #1: Partly

Reviewer #2: Yes

Reviewer #3: Yes

Reviewer #4: Partly

2. Has the statistical analysis been performed appropriately and rigorously? 

Reviewer #1: Yes

Reviewer #2: Yes

Reviewer #3: Yes

Reviewer #4: Yes

3. Have the authors made all data underlying the findings in their manuscript fully available?

Reviewer #1: No

Reviewer #2: Yes

Reviewer #3: Yes

Reviewer #4: Yes

4. Is the manuscript presented in an intelligible fashion and written in standard English?

Reviewer #1: Yes

Reviewer #2: Yes

Reviewer #3: Yes

Reviewer #4: Yes

5. Review Comments to the Author

Reviewer #1: THE STUDY IS VERY RELEVANT FOR GERIATRIC ORTHOPAEDIC PRACTICE.

THE CLASSIFICATION OF THE STUDY VARIABLES IS QUESTIONABLE AND UNCLEAR . IN THIS AGE GROUP URINARY TRACT INFECTION CAN BE LIIFE THREATENING AND IT CAN BE VERY SEVERE.

THE MAIN INDEPENDENT VARIABLES ARE VERY GOOD BUT ARE THEY CLINICAL OR HOSPITAL FACTORS?

THE LABORATORY FACTORS WERE BROAD AND SOME OF THEM ARE THE SAME WITH THE CLINICAL FACTORS E.G CHRONIC KIDNEY DISEASE AND UREA,ELECTROLYTE AND CREATINE.

THE STUDY WILL HELP IMPROVE PATIENT EVALUATION FOR HIP SURGERIES TO POSSIBLY BETTER THE OUTCOME.

THE DOCUMENTED TITLE THOUGH IS NON-SPECIFIC AND BROAD WAS PARTLY ANSWERED IN THE CONCLUSION

I WAS UNABLE TO ACCESS THE DOCUMENTS ATTACHED (HENCE QUESTION 3 RESPONSE IS I DON'T KNOW)

Reviewer #2: The manuscript presents an observational retrospective cohort study that aims to identify clinical, laboratory, and hospital factors associated with preoperative complications in older adults with hip fractures. Overall, the quality of the writing and the study design is sound, and the article is suitable for acceptance. Below are the strengths of the manuscript and a few minor suggestions for improvement:

Clarification of Terms: The term "mental red cross >=3 points" could be clarified or defined in more detail, as it may not be a widely recognized metric outside specific regional or institutional practices.

Further Discussion on Gender Differences: While the manuscript notes that female gender is associated with an increased risk of preoperative complications, a deeper exploration of potential underlying causes or mechanisms would enhance the discussion section.

Minor Grammatical Revisions: Although the language is generally good, a few sentences could benefit from minor grammatical refinement for smoother readability.

Conclusion:

The manuscript is well-structured and provides important insights into preoperative complications in older adults with hip fractures. With minor revisions, I recommend this manuscript for acceptance.

Reviewer #3: Question 1: The introduction of the article is more detailed and provides relevant background information. However, its structure can be somewhat simplified. Merge similar content and consider breaking it into fewer paragraphs.

Question 2: On page 5, line 109, "2.3 Sampling and sampling" why is it so stated

Question 3: Too few variables were included in the multivariate analysis compared to the baseline data, and there may be large confounding factors. It is necessary to consider the collinearity problem, but the criterion of excluding collinearity by P < 0.2 is there any relevant literature report, please give the specific VIF value of each variable.

Question 4: The points described in the discussion part of the article are more specific, but they are not in-depth. I hope to see the author's in-depth analysis of the main conclusions of the article on the mechanism level.

Reviewer #4: The sample size calculation, although mentioned as being performed using the Cleveland Clinic online calculator, lacks detailed explanation of the parameter selection. More information should be provided on the sample size determination process, and the rationale behind the choice of parameters should be clarified, along with how the representativeness of the sample was ensured. Additionally, the use of non-random sampling limits the generalizability of the findings, which should be further emphasized in the discussion.

The exclusion criteria are not sufficiently justified. For instance, patients with voluntary discharge are excluded without discussion of whether they differ significantly from included patients in terms of health status or outcomes, potentially introducing bias. It is recommended to provide a more detailed rationale for excluding these patients and to discuss how their exclusion might impact the study’s results.

The data collection period spans from 2017 to 2019, but there is no mention of whether any changes in hospital practices occurred during this time and how these changes might have influenced the results. Given the retrospective nature of the study, the potential influence of time-related factors should be considered and discussed.

The writing style is somewhat verbose, particularly in the abstract and discussion sections. Some paragraphs could be simplified to improve clarity and conciseness. In particular, the introduction contains excessive background information that could be streamlined to focus more on the research question.

Inconsistent terminology usage is evident throughout the manuscript, particularly with the term "mental red cross," which is not clearly defined and is used inconsistently. It is recommended to ensure consistent use of terminology and to provide a detailed explanation of unfamiliar terms upon first use to avoid confusion.

Some statements in the manuscript lack precision. For example, in the multivariate analysis results, the assertion that "female sex significantly increases the risk of pre-surgical complications" is overconfident given that the p-value is 0.052, which is marginally significant. It is advised to use more cautious phrasing, such as "female sex showed a trend towards an association with pre-surgical complications."

The discussion of certain variables, such as cardiovascular disease and diabetes, lacks depth. The manuscript would benefit from a more thorough exploration of the potential mechanisms through which these variables might influence preoperative complications, adding credibility to the findings.

The use of a Poisson regression model for multivariate analysis is reasonable, but the choice of this model over others, such as logistic regression, is not explained. A brief justification of the model selection should be provided, and the handling of multicollinearity among variables should be discussed.

Confidence intervals are missing for some variables, especially in Table 3 of the multivariate regression analysis. The absence of confidence intervals limits the ability to assess the precision of the findings. It is recommended to include 95% confidence intervals for all relative risk estimates.

Some effect sizes, while statistically significant, may lack clinical significance. For instance, the effect of admission glucose levels on pre-surgical complications (RR=1.01) is statistically significant but the magnitude of the effect is small. Further discussion is needed to clarify the clinical relevance of such findings and to avoid overinterpretation of statistically significant results with minimal effect sizes.

Missing data are acknowledged but not adequately addressed. While the manuscript notes that some variables, such as sex and Barthel Index, have missing values, it does not describe how these missing data were handled (e.g., exclusion, imputation). A more detailed explanation of the approach to missing data is necessary.

The presentation of data in Table 1 is too condensed, with an overwhelming amount of information. It is recommended to split the table into multiple sections—one for demographic and clinical characteristics, and another for laboratory and complication data—to enhance readability. Additionally, the inclusion of graphical representations, such as Kaplan-Meier curves for complication occurrence, would improve the accessibility of the findings.

The structure of the discussion is somewhat disjointed, with weak logical connections between certain points. The comparison of study results with existing literature is brief, and the manuscript would benefit from a more structured and thorough analysis of how the findings align with or differ from previous studies.

The ethics section mentions approval from an ethics committee, but more detailed information on how data privacy was ensured during data collection and analysis is needed, especially given the use of personal health data. Additional information on how patient anonymity was maintained would strengthen the ethical compliance of the study.

The study claims an informed consent waiver due to its retrospective design, but more justification is needed for this waiver, including specific references to relevant ethical guidelines. Further elaboration on the justification for waiving informed consent and how the study ensured compliance with ethical standards would enhance the credibility of the ethical considerations.

6. PLOS authors have the option to publish the peer review history of their article (what does this mean?). If published, this will include your full peer review and any attached files.

Reviewer #1: **Yes: **Mustapha ALIMI

Orthopaedic and Spine Surgeon

National Orthopaedic Hospital

Igbobi ,Lagos

Nigeria

Reviewer #2: No

Reviewer #3: No

Reviewer #4: No

---

## [Author Response · Author response to Decision Letter 0]

13 Oct 2024

Dear Editor,

Thank you very much for reviewing our article, "Clinical, laboratory, and hospital factors associated with preoperative complications in Peruvian older adults with hip fracture". Your suggestions and comments will be addressed below. Thank you for your valuable time and excellent review.

Editor's comments

A well written article which requires clarification of the methodology and a broader overview of the literature including alternate means of risk stratification peri-operatively.

Our response: 

The methodology was clarified according to the observations of the other reviewers in order to provide greater precision to the findings; furthermore, these have been reviewed more thoroughly based on the literature found, considering that there are few studies reporting presurgical complications and that they focus more on postsurgical complications in older adult patients with hip fractures. Thank you for your comment.

The grammar of the English text was revised. The article was corrected according to each of the reviewers' comments. References were updated to align them with the objectives and results of the study. In addition, the scientific names and affiliations of the authors were verified.

Reviewer #1: 

1. Reviewer says: THE STUDY IS VERY RELEVANT FOR GERIATRIC ORTHOPAEDIC PRACTICE.

Our response: 

Thank you for your comment.

2. Reviewer says: THE CLASSIFICATION OF THE STUDY VARIABLES IS QUESTIONABLE AND UNCLEAR. IN THIS AGE GROUP URINARY TRACT INFECTION CAN BE LIIFE THREATENING AND IT CAN BE VERY SEVERE.

Our response: 

Complications were classified according to their risk status as minor and major. Sepsis, which is assumed to be of urinary or other origin, is considered major, while urinary tract infection is considered minor if it does not meet sepsis criteria. We rely on a classification of the reference number 18. Added in the definition of the variable. 

3. Reviewer says: THE MAIN INDEPENDENT VARIABLES ARE VERY GOOD BUT ARE THEY CLINICAL OR HOSPITAL FACTORS?

Our response: 

Emergency stay time and preoperative time are considered hospital factors, and the rest involve clinical and laboratory characteristics.

4. Reviewer says: THE LABORATORY FACTORS WERE BROAD AND SOME OF THEM ARE THE SAME WITH THE CLINICAL FACTORS E.G CHRONIC KIDNEY DISEASE AND UREA,ELECTROLYTE AND CREATINE.

Our response: 

Urea and creatinine are considered analytical values that could be associated with an acute, chronic, or exacerbated chronic condition and were therefore measured as a numerical value independently of the antecedent. This is because in a hip fracture there is a high probability of blood loss with consequent renal injury, which in other literature is considered a major complication. We did not measure electrolytes.

5. Reviewer says: THE STUDY WILL HELP IMPROVE PATIENT EVALUATION FOR HIP SURGERIES TO POSSIBLY BETTER THE OUTCOME.

Our response: 

Thank you for your comment.

6. Reviewer says: THE DOCUMENTED TITLE THOUGH IS NON-SPECIFIC AND BROAD WAS PARTLY ANSWERED IN THE CONCLUSION

Our response: 

Our study does not evaluate a particular association with one variable but explores associations with multiple variables due to the paucity of evidence from studies for pre-surgical complications. The conclusions describe the most frequent complications, and the factors associated with the risk of developing them. Thank you for your comment

7. Reviewer says: I WAS UNABLE TO ACCESS THE DOCUMENTS ATTACHED (HENCE QUESTION 3 RESPONSE IS I DON'T KNOW)

Our response: 

 The database has been uploaded to the figshare website 

http://doi.org/10.6084/m9.figshare.26507296.v1. 

 All other reviewers have managed to access the files, thank you for your comments

Reviewer #2: 

1. Reviewer says: The manuscript presents an observational retrospective cohort study that aims to identify clinical, laboratory, and hospital factors associated with preoperative complications in older adults with hip fractures. Overall, the quality of the writing and the study design is sound, and the article is suitable for acceptance. Below are the strengths of the manuscript and a few minor suggestions for improvement:

Our response: 

Thank you for your comment.

2. Reviewer says: Clarification of Terms: The term "mental red cross >=3 points" could be clarified or defined in more detail, as it may not be a widely recognized metric outside specific regional or institutional practices.

Our response: 

The variable Mental Red Cross is defined in material and method, but a major reference is added.

3. Reviewer says: Further Discussion on Gender Differences: While the manuscript notes that female gender is associated with an increased risk of preoperative complications, a deeper exploration of potential underlying causes or mechanisms would enhance the discussion section.

Our response: 

 The differences in gender are highlighted in the discussion, along with possible

 explanations for these findings.

4. Reviewer says: Minor Grammatical Revisions: Although the language is generally good, a few sentences could benefit from minor grammatical refinement for smoother readability.

Our response: 

Revised grammar.

5. Reviewer says: Conclusion:

The manuscript is well-structured and provides important insights into preoperative complications in older adults with hip fractures. With minor revisions, I recommend this manuscript for acceptance.

Our response: 

 Thank you for your comment.

Reviewer #3: 

1. Reviewer says: Question 1: The introduction of the article is more detailed and provides relevant background information. However, its structure can be somewhat simplified. Merge similar content and consider breaking it into fewer paragraphs.

Our response: 

Paragraphs summarizing data have been joined together. It is important to note that the introduction attempts to describe the condition, then describes the problem situation with literature, identifies an information niche, and finally describes the objective and importance of the study.

2. Reviewer says: Question 2: On page 5, line 109, "2.3 Sampling and sampling" why is it so stated

Our response: 

 It was modified.

3. Reviewer says: Question 3: Too few variables were included in the multivariate analysis compared to the baseline data, and there may be large confounding factors. It is necessary to consider the collinearity problem, but the criterion of excluding collinearity by P < 0.2 is there any relevant literature report, please give the specific VIF value of each variable.

Our response: 

 The robust and adjusted analysis was detailed, and the reason for the inclusion of certain

 variables was specified step by step; additionally, the VIF values were included.

4. Reviewer says: Question 4: The points described in the discussion part of the article are more specific, but they are not in-depth. I hope to see the author's in-depth analysis of the main conclusions of the article on the mechanism level.

Our response: 

 The main findings are detailed, making a greater comparison with other studies and explaining possible differences.

Reviewer #4: 

1. Reviewer says: The sample size calculation, although mentioned as being performed using the Cleveland Clinic online calculator, lacks detailed explanation of the parameter selection. More information should be provided on the sample size determination process, and the rationale behind the choice of parameters should be clarified, along with how the representativeness of the sample was ensured. Additionally, the use of non-random sampling limits the generalizability of the findings, which should be further emphasized in the discussion.

Our response: 

It was modified in the section on sample and sampling.

The implications of non-probability sampling are specified in limitations.

2. Reviewer says: The exclusion criteria are not sufficiently justified. For instance, patients with voluntary discharge are excluded without discussion of whether they differ significantly from included patients in terms of health status or outcomes, potentially introducing bias. It is recommended to provide a more detailed rationale for excluding these patients and to discuss how their exclusion might impact the study’s results.

Our response: 

It is detailed in the figure of results the patients excluded according to criteria, and in the methods section, it is described that this exclusion could not affect the analysis of the results and that it is not necessary to conduct an analysis of that subpopulation.

3. Reviewer says: The data collection period spans from 2017 to 2019, but there is no mention of whether any changes in hospital practices occurred during this time and how these changes might have influenced the results. Given the retrospective nature of the study, the potential influence of time-related factors should be considered and discussed.

Our response: 

 It is described in the strengths of the study that the data was provided by a geriatric 

 orthopedics team that was established in the year the information collection began and

 that the measurement of outcomes was conducted during the patient's hospital stay.

4. Reviewer says: The writing style is somewhat verbose, particularly in the abstract and discussion sections. Some paragraphs could be simplified to improve clarity and conciseness. In particular, the introduction contains excessive background information that could be streamlined to focus more on the research question.

Our response: 

 This was modified in the summary and introduction.

5. Reviewer says: Inconsistent terminology usage is evident throughout the manuscript, particularly with the term "mental red cross," which is not clearly defined and is used inconsistently. It is recommended to ensure consistent use of terminology and to provide a detailed explanation of unfamiliar terms upon first use to avoid confusion.

Our response: 

The mental red cross test is defined in the variables section of the methodology, and the term test is added in the rest of the manuscript.

6. Reviewer says: Some statements in the manuscript lack precision. For example, in the multivariate analysis results, the assertion that "female sex significantly increases the risk of pre-surgical complications" is overconfident given that the p-value is 0.052, which is marginally significant. It is advised to use more cautious phrasing, such as "female sex showed a trend towards an association with pre-surgical complications."

Our response: 

 It was modified in results, discussion, and conclusions.

7. Reviewer says: The discussion of certain variables, such as cardiovascular disease and diabetes, lacks depth. The manuscript would benefit from a more thorough exploration of the potential mechanisms through which these variables might influence preoperative complications, adding credibility to the findings.

Our response: 

 Pathologies such as diabetes or cardiovascular disease are not detailed because the

 findings explain an association; however, multimorbidity is explored in depth as an

 entity.

8. Reviewer says: The use of a Poisson regression model for multivariate analysis is reasonable, but the choice of this model over others, such as logistic regression, is not explained. A brief justification of the model selection should be provided, and the handling of multicollinearity among variables should be discussed.

Our response: 

The explanation of the model and the analysis of multicollinearity are described in the multivariate analysis section.

9. Reviewer says: Confidence intervals are missing for some variables, especially in Table 3 of the multivariate regression analysis. The absence of confidence intervals limits the ability to assess the precision of the findings. It is recommended to include 95% confidence intervals for all relative risk estimates.

Our response: 

The absence of confidence intervals in some variables in the adjusted analysis column is due to the fact that these variables were not included in the analysis, which is explained in the results section of the multivariate analysis.

10. Reviewer says: Some effect sizes, while statistically significant, may lack clinical significance. For instance, the effect of admission glucose levels on pre-surgical complications (RR=1.01) is statistically significant but the magnitude of the effect is small. Further discussion is needed to clarify the clinical relevance of such findings and to avoid overinterpretation of statistically significant results with minimal effect sizes.

Our response: 

A detailed analysis was conducted on the relevance of the association between glucose levels and preoperative complications in the discussion.

11. Reviewer says: Missing data are acknowledged but not adequately addressed. While the manuscript notes that some variables, such as sex and Barthel Index, have missing values, it does not describe how these missing data were handled (e.g., exclusion, imputation). A more detailed explanation of the approach to missing data is necessary.

Our response: 

The handling of missing data is detailed in the results.

12. Reviewer says: The presentation of data in Table 1 is too condensed, with an overwhelming amount of information. It is recommended to split the table into multiple sections—one for demographic and clinical characteristics, and another for laboratory and complication data—to enhance readability. Additionally, the inclusion of graphical representations, such as Kaplan-Meier curves for complication occurrence, would improve the accessibility of the findings.

Our response: 

 Table 1 allows us to condense the information on general characteristics, which include clinical, laboratory, and hospital aspects. Considering that these last ones are few variables, we do not consider adding an additional table. The survival graphs are not related to the study's objective, and we do not consider them relevant. Thank you for the comments.

13. Reviewer says: The structure of the discussion is somewhat disjointed, with weak logical connections between certain points. The comparison of study results with existing literature is brief, and the manuscript would benefit from a more structured and thorough analysis of how the findings align with or differ from previous studies.

Our response: 

 There is a depth in the main findings, making a respective comparison and analyzing

 these findings. There is already a structure that has been respected to emphasize the

 findings.

14. Reviewer says: The ethics section mentions approval from an ethics committee, but more detailed information on how data privacy was ensured during data collection and analysis is needed, especially given the use of personal health data. Additional information on how patient anonymity was maintained would strengthen the ethical compliance of the study.

Our response: 

The process of data collection and the coding of the identifications of the study population were specified.

15. Reviewer says: The study claims an informed consent waiver due to its retrospective design, but more justification is needed for this waiver, including specific references to relevant ethical guidelines. Further elaboration on the justification for waiving informed consent and how the study ensured compliance with ethical standards would enhance the credibility of the ethical considerations.

Our response: 

 The justification was modified, and a reference was added.

Sincerely, 

Mario J. Valladares-Garrido 

Universidad Continental, Lima 15046, Peru; mvalladares@continental.edu.pe

---

## [Editor Report · Decision Letter 1]

18 Oct 2024

Clinical, laboratory, and hospital factors associated with preoperative complications in Peruvian older adults with hip fracture.

PONE-D-24-33388R1

Dear Dr. Valladares-Garrido,

We’re pleased to inform you that your manuscript has been judged scientifically suitable for publication and will be formally accepted for publication once it meets all outstanding technical requirements.

Kind regards,

Barry Kweh

Academic Editor

PLOS ONE

Additional Editor Comments (optional):

The authors have satisfactorily clarified the definition of their variables and strengthened their discussion with a broader incorporation of the literature regarding perioperative complications of patients undergoing hip fractures.

---

## [Editor Report · Acceptance letter]

24 Oct 2024

PONE-D-24-33388R1 

PLOS ONE

Dear Dr. Valladares-Garrido, 

I'm pleased to inform you that your manuscript has been deemed suitable for publication in PLOS ONE. Congratulations! Your manuscript is now being handed over to our production team.

Kind regards, 

on behalf of

Dr. Barry Kweh 

Academic Editor

PLOS ONE